# Differential effects of COVID-19 and containment measures on mental health: Evidence from ITA.LI—Italian Lives, the Italian household panel

**Mario Lucchini[1], Tiziano Gerosa[1], Marta Pancheva[2], Maurizio Pisati[1], Chiara Respi[1], Egidio Riva[1]***

**1** Department of Sociology and Social Research, University of Milano-Bicocca, Milan, Italy, **2** Department of Economics and Management, Sophia University Institute, Figline e Incisa Valdarno, Italy

\* egidio.riva@unimib.it

## Abstract

This study used a subsample of a household panel study in Italy to track changes in mental health before the onset of COVID-19 and into the first lockdown period, from late April to early September 2020. The results of the random-effects regression analyses fitted on a sample of respondents aged 16 years and older (N = 897) proved that there was a substantial and statistically significant short-term deterioration in mental health (from 78,5 to 67,9; β = -10.5, p < .001; Cohen's d -.445), as measured by a composite index derived from the mental component of the 12-item Short-Form Health Survey (SF-12). The findings also showed heterogeneity in the COVID-related effects. On the one hand, evidence has emerged that the pandemic acted as a great leveller of pre-existing differences in mental health across people of different ages: the decrease was most pronounced among those aged 16–34 (from 84,2 to 66,5; β = -17.7, p < .001; Cohen's d -.744); however, the magnitude of change reduced as age increased and turned to be non-significant among individuals aged 70 and over. On the other hand, the COVID-19 emergency widened the mental health gender gap and created new inequalities, based on the age of the youngest child being taken care of within the household.

## Introduction

The ongoing SARS-CoV-2 (hereafter COVID-19) pandemic is unprecedented when compared to earlier periods of adversity. Leading to the most severe economic shock the world has experienced in decades, it has triggered a global emergency, the course of which has brought about rapid changes to people's everyday lives, and it is likely to have both short and long-lasting consequences. A mounting body of evidence produced internationally indicates that mental health has significantly declined since the COVID-19 outbreak. Indeed, increased levels of psychological distress—in the form of anxiety, depressive symptoms, stress, fear of death, and insomnia—have been registered in both cross-sectional [1–14] and longitudinal studies [15–21]. Socio-demographic characteristics, household structure and composition, employment status and financial strain, underlying health conditions, and the living space available or

**Data Availability Statement:** In accordance with EU General Data Protection Regulation (EU 2016/79), University of Milano-Bicocca internal

regulations (D.R. 6256/2018, prot. 90980/18) ITA. LI – Italian Lives data are fully encrypted, stored anonymously in the cloud, and protected against unauthorized access, disclosure, modification, or destruction. ITA.LI – Italian Lives data (specifically ITA.LI – Italian Lives wave 1 and ITA.LI COVID-19 survey data, which have been used for this study) are currently available only to researchers working at the ITA.LI – Italian Lives project who completed a registration process at the personal data protection office, in accordance with the above-mentioned regulations. As already stated at the time of first submission, the same data will be publicly available to researchers from University of Milano-Bicocca and through Cross National Equivalent File (https://www.cnefdata.org/) in due time. However, the data release policy (time of public release, details of how to apply for and access the datasets, end user licence, etc.) has not been formally defined yet. Due to this lack of formal policies and procedures, all data underlying the findings may be currently accessed, only for the purpose of reproducing the analyses, through the corresponding author (or any of the remaining authors) and the personal data protection officer at University of Milano-Bicocca. The personal data protection officer can be contacted (at rpd@unimib.it or certified email rpd@pec.unimib.it) for all queries concerning personal data processing and the exercise of any rights deriving from General Data Protection Regulation (EU 2016/79). ITA.LI – Italian Lives Data Controller is the University of Milano-Bicocca, represented by its legal representative, the Rector Giovanna Iannantuoni (rettorato@unimib.it or or certified email ateneo.bicocca@pec.unimib.it). All relevant materials that may be reasonably requested by others to reproduce the results will made available upon the publication of the study.

**Funding:** ITA.LI – Italian Lives project is funded by the Italian Ministry of Education, Universities and Research under the "Departments of Excellence 2018-2022" initiative (Italian Law 232 of 11 December 2016) (https://www.miur.gov.it/dipartimenti-di-eccellenza). Internal grant number at the Department of Sociology and Social Research of the University of Milan-Bicocca is 2018-NAZ-0116. The award was received by the Department of Sociology and Social Research of the University of Milan-Bicocca. The funders had no role in study design, data collection and analysis, decision to publish, or preparation of the manuscript.

**Competing interests:** The authors have declared that no competing interests exist.

people's satisfaction with their housing are among the factors that can play a crucial role in increasing or mitigating the COVID-related consequences for mental health. For instance, recent research suggests that women and young adults have been negatively affected [3–6, 10–17]. Individuals living with children or in an overcrowded house, as well as those living alone, have reported worse mental health conditions [2, 15, 16]. Unemployment or disrupted working activity due to lockdown or containment measures [6, 11, 15], as well as a lower position in income distribution [9, 15], have also had a deteriorating psychological effect. Finally, direct or indirect exposure to COVID-19 has been proven to be a predictor of a higher impact of the outbreak on depression and anxiety [3–5, 16, 17].

In countries such as Italy, the nature and magnitude of the possible psychosocial consequences of COVID-19 are difficult to assess. The empirical evidence available is anecdotal or cross-sectional [e.g. 4–6]. Hence, longitudinal studies are needed to address the individual unobserved heterogeneity problem and the problem of reversed causality, while shedding light on the cumulative nature of life courses—that is on how the previous life experiences are linked to subsequent experiences—during the pandemic and beyond. Otherwise, it remains unclear whether and, if so, to what extent the COVID-19 outbreak has been worsening or reducing existing inequalities or even creating new ones in relation to mental health [22].

Against this background, this paper investigates the short-term COVID-related consequences on mental health—as assessed by a composite index derived from the mental component of the 12-item Short-Form Health Survey (SF-12) [23]—on a longitudinal sample (N = 897) of individuals (aged 16+) in Italy. Drawing on available evidence on COVID-19 [24] and previous health or economic crises [25–27], we expect to find a generalised deterioration of mental health (Hypothesis 1). In addition, building on longitudinal studies conducted at times of COVID-19 in countries such as the UK and Switzerland [15–17, 28], we may anticipate heterogeneity in the pandemic-related effects on mental health. Specifically, following Kuhn and colleagues [29] we hypothesised a stronger deterioration of mental health among specific subgroups such as *a)* those more exposed to the detrimental effects of social isolation, such as young adults, individuals living without a partner, and people belonging to a COVID-19 risk group (i.e. those tested because had symptoms or were potentially exposed to the virus and people living in municipalities with high rates of new coronavirus infections) (Hypothesis 2); *b)* individuals with a heavier workload, such as women and people with preschool-age children in the household (Hypothesis 3); and *c)* individuals with fewer socioeconomic resources, such as the lower educated, the unemployed, and people with poorer housing conditions (Hypothesis 4). Indeed, there is reason to believe that the COVID-19 emergency may expose the more vulnerable groups and exacerbate mental health inequalities; however, it may also act as a great leveller and reduce gaps in mental health existing before the epidemic, to the extent that all people, not just the more disadvantaged, struggle to cope with the new circumstances [15].

## Data and methods

### Data source and sample

Our analyses draw on data obtained from an ad-hoc survey on the impact of COVID-19 on individuals' everyday lives in Italy (ITA.LI COVID-19), which was conducted on a sample of respondents to ITA.LI—Italian Lives (ITA.LI). ITA.LI is a newly established longitudinal study on a probability sample of 4,900 households and 8,967 individuals (aged 16+) living in 280 municipalities in Italy. The first wave of data collection started in June 2019 and finished in December 2020, gathering information on a broad range of topics, such as education, employment and working conditions, family life and caring, wealth, health, well-being, and housing and residential mobility. From 20 April 2020, all panel members who had already

taken part in ITA.LI wave 1 were invited, using SMS or email as contact modes, to answer the ITA.LI COVID-19 survey using computer-aided web interviewing (CAWI) and computer-assisted telephone interviewing (CATI) methods [30]. The questionnaire collected information on the consequences of the pandemic—mostly through measures that were already used in ITA.LI wave 1—on the quality of life, health and well-being, employment status and working conditions, family and social relationships, children, and distance education. Moreover, the ITA.LI COVID-19 survey included specific questions on health issues such as perception of the risk of infection, preventive behaviours related to the pandemic, testing for coronavirus, and self-isolation. Overall, 950 of 2,415 eligible people (i.e. respondents who took part in ITA.LI wave 1 and for whom an email and/or a phone number was/ were available and valid) participated in the ITA.LI COVID-19 survey [30], which ended on 2 September 2020. The final response rate was 39.3% (AAPOR response rate 1). When merging data collected from the records of the respondents to ITA.LI wave 1, 53 participants were excluded because they were unmatched to previous interviews or because they completed the questionnaire after 9 March 2020. The final sample (Table 1) comprised 897 respondents. Observations containing missing values on any of the variables included in the models were omitted from analysis.

## Variables

**Dependent variable: Mental health.** A summary measure of mental health was constructed using the following six items, which are generally employed to assess the mental component of the SF-12 [18]: 'During the past 4 weeks. . .' 1) 'have you accomplished less than you would have liked?'; 2) 'did you fail to do work or other activities as carefully as usual?'; 3) 'have you felt calm and peaceful?'; 4) 'did you have a lot of energy?'; 5) 'have you felt downhearted and low?'; and 6) 'has your health limited your social activities?'. Following SF-12 guidelines, items were recoded where necessary so that higher scores indicated better mental health. Specifically, items 1 and 2 were dummy-coded (yes or no), while items 3 and 4 were coded into six categories ranging from 0 (none of the time) to 5 (all of the time). Item 5 was coded into six categories ranging from 0 (all of the time) to 5 (none of the time). Item 6 was coded into five categories ranging from 0 (all of the time) to 4 (none of the time). The dimensionality of the single-item summary measure of mental health was assessed over time using pooled data, that is, data from the ITA.LI wave 1 and ITA.LI COVID-19 survey [21]. The suitability of the data was first assessed by analysing the determinant of the correlation matrix (det = .112), Kaiser-Meyer-Olkin measure of sampling adequacy (KMO = .784), and Bartlett's test of sphericity ($\chi2$ = 3838.8, df = 15, p < .001). Subsequently, principal component analysis was performed. The first component or factor, which had the largest eigenvalue (3.190) and explained 53% of the total variance, was retained. Factor loadings ranged from .697 for Item 5 to .753 for Item 6 (Table 2). On the basis of the Eigenvalues-greater-than-one rule, only the first component or factor appeared to be meaningful; thus, the unidimensionality of the item response data could be detected. This assessment was confirmed by Horn parallel analysis with 5,000 iterations, which indicated that one factor was retained, with an adjusted eigenvalue of 3.121. The internal consistency of the scale formed from the six items was tested and assessed by computing Cronbach's alpha (.778). The mental health factor was scored on the entire sample for both measurement occasions using the regression method, and the values were normalised within a 0–100 range (Fig 1). Finally, the convergent validity of the resulting measure of mental health was assessed by estimating its correlation with the mental component of the SF-12, which was computed from ITA.LI wave 1 data using the standard US algorithm. The Pearson product-moment correlation coefficient (.958, p< .001) indicated that, although measured in different

**Table 1. Sample.**

| | ITA.LI wave 1 (Pre-lockdown) | | | | ITA.LI COVID-19 (Post-lockdown) | | | |
|---|---|---|---|---|---|---|---|---|
| | **M** | **SD** | **N** | **(%)** | **M** | **SD** | **N** | **(%)** |
| Mental health | 78.5 | 17.7 | 875 | 97.6 | 67.9 | 22.2 | 880 | 98.1 |
| *Missing* | | | *22* | *2.4* | | | *17* | *1.9* |
| Age | | | | | | | | |
| 16–34 | | | | | | | 149 | 16.6 |
| 35–44 | | | | | | | 142 | 15.8 |
| 45–54 | | | | | | | 203 | 22.6 |
| 55–69 | | | | | | | 268 | 29.9 |
| 70 or more | | | | | | | 135 | 15.1 |
| Living with a partner | | | | | | | | |
| No | | | | | | | 384 | 42.8 |
| Yes | | | | | | | 513 | 57.2 |
| Testing for COVID–19 | | | | | | | | |
| No | | | | | | | 803 | 89.6 |
| Yes | | | | | | | 93 | 10.4 |
| *Missing* | | | | | | | *1* | |
| Increase in mortality rate at the municipal level | | | | | | | | |
| Up to 10% | | | | | | | 482 | 53.7 |
| 11–50% | | | | | | | 316 | 35.2 |
| 51% or more | | | | | | | 99 | 11.1 |
| Sex | | | | | | | | |
| Male | | | | | | | 357 | 39.8 |
| Female | | | | | | | 540 | 60.2 |
| Age of the youngest child | | | | | | | | |
| No child 0–14 years | | | | | | | 741 | 82.6 |
| 0–6 years | | | | | | | 74 | 8.3 |
| 7–14 years | | | | | | | 82 | 9.1 |
| Education˚ | | | | | | | | |
| Up to lower secondary | | | 277 | 30.9 | | | | |
| Upper secondary | | | 459 | 51.2 | | | | |
| Tertiary | | | 161 | 18.0 | | | | |
| Employment status˚ | | | | | | | | |
| Employed | | | 452 | 50.4 | | | | |
| Unemployed | | | 66 | 7.4 | | | | |
| Economically inactive | | | 158 | 17.6 | | | | |
| Retired | | | 221 | 24.6 | | | | |
| Shortage of living space˚ | | | | | | | | |
| No | | | 792 | 89.9 | | | | |
| Yes | | | 89 | 10.1 | | | | |
| *Missing* | | | *16* | | | | | |

˚ Variables measured in ITA.LI wave 1.

ways, the mental health factor and the mental component of the SF-12 were strongly correlated.

**Independent variables.** Building on recent research, as well as on studies on earlier pandemics and economic crises [24, 31–33], a set of independent variables were included in the

**Table 2. Weights (principal components loadings) and internal reliability of the mental health summary measure.**

| Item | Standardized factor loadings | Scale reliability coefficient |
|---|---|---|
| I1—Accomplished less | 0.750 | |
| I2—Do activities less carefully | 0.743 | |
| I3—Feeling calm and peaceful* | 0.701 | |
| I4—Having a lot of energy* | 0.731 | |
| I5—Feeling downhearted and low | 0.697 | |
| I6—Limited social activities | 0.751 | |
| Total scale | | 0.778 |

* Reverse-coded survey items.

models. To test the social isolation hypothesis, we included age (recoded into five categories: 16–34, 35–44, 45–54, 55–69, and 70+), an item assessing whether the respondent lived with a partner (yes or no), and two items measuring respondents' direct and indirect exposure to COVID-19. The first item measured whether the respondent had taken either a serological or nasopharyngeal swab test (yes or no). The second item assessed the mortality risk of COVID-19 at the local level. More specifically, we extracted mortality statistics, namely, monthly death registration data, at the municipal level for six years (2015 to 2020), and calculated changes in monthly mortality rates by comparing the 2020 average (from January to the month of the interview) with the five-year (2015 to 2019) average (from January to the month of the

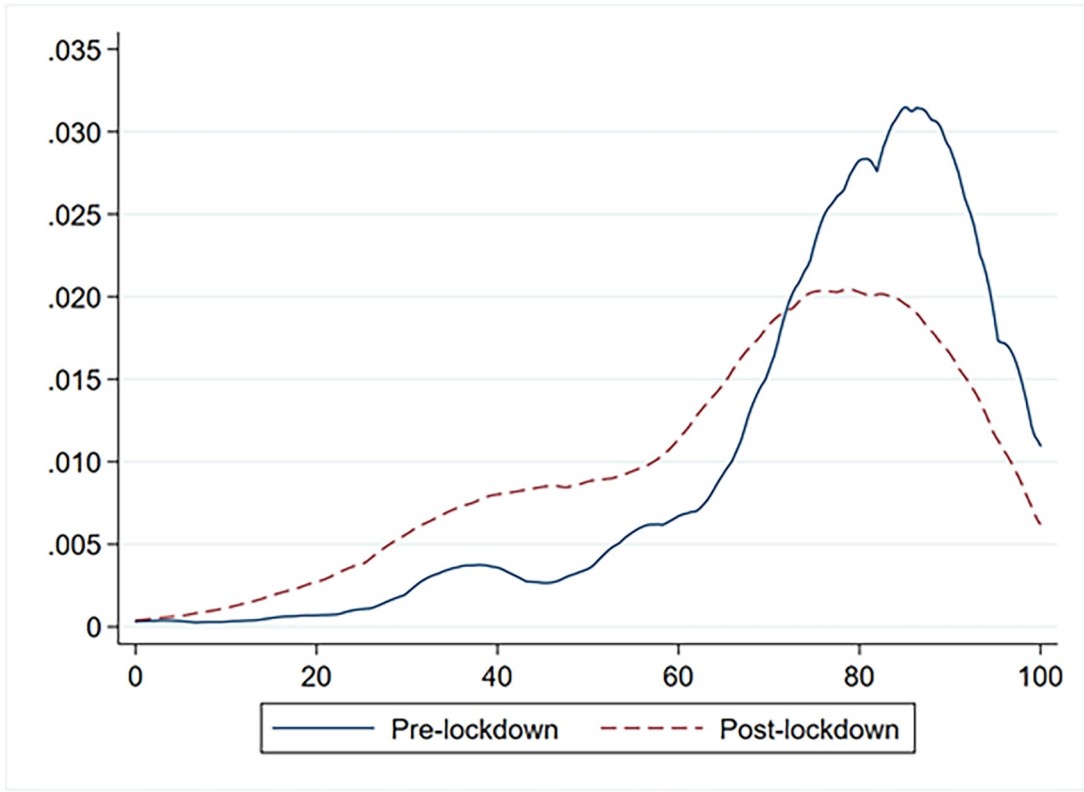

**Fig 1. Kernel distribution of the pre- and post-Covid mental health latent-trait scores.**

interview). The resulting variable, which measured changes or differences in mortality rates at the municipal level, was coded into three categories (less than 10% increase, 11% to 50% increase, more than 50% increase) and merged with data on respondents based on their municipality of residence. For the workload hypothesis, we inserted respondents' sex into the model and the age of the youngest child living in the same household and taken care of by the respondent (no child aged 0–14 years; 0–6 years; 7–14 years). Socioeconomic variables encompassed employment status (coded into four categories: employed, unemployed, economically inactive—i.e. persons of working-age outside the labour force, such as students and house-wives—and retired), education (coded into three categories: up to lower secondary, upper secondary, and tertiary), and the respondent's perception of the shortage of living space (yes or no). Values of the socioeconomic variables were extracted from ITA.LI wave 1.

## Analytic strategy

The impact of COVID-19 on mental health was investigated using a one-group pre-test–post-test design. The random-effects (RE) estimator was preferred, as more appropriate, over the fixed-effects (FE) estimator of the model parameters on the basis of the results of the Hausman Test (chi2 = 24.35; prob>chi2 = 0.14). Following recent studies on the consequences of the pandemic [e.g. 15], we first estimated a RE model with pre- and post-lockdown period indicator as the only predictor of variation in mental health. Subsequently, the interactions between the pre- and post-lockdown period indicator and the entire set of explanatory variables were included. The parameter estimates of this multiple-interaction model are interpretable as the change in mental health scores within a specific subgroup. For each of the two models we computed both the unadjusted p-values of the estimates and Cohen's d; the latter is an effect size measure for single group pre-post study designs, which assesses the magnitude of changes in mental health scores overtime calculating the difference between the post- and the pre-test means and dividing such difference by the standard deviations of the differences. In addition, we ran the Wald test to perform some joint tests of group comparisons, that is, to estimate differences in the COVID-related effects on mental health across the levels of independent variables. The Bonferroni method of correcting p-values was used in Model 2 to counteract the problem of multiple testing. Given that ITA.LI wave 1 data used for this study were collected from June 2019 to March 2020, whereas the ITA.LI COVID-19 survey was conducted over four months in 2020, we carried out a sensitivity analysis to check for potential seasonality effects. To do so, we replicated the RE regression models on the subsample of respondents (N = 257) who took part in ITA.LI wave 1 over the same four months. The results of this additional set of analyses are reported in Table 1 in S1 File. Analyses were performed using STATA 17.

## Findings

Table 3 displays the results of the RE regression models that were fitted to estimate both changes in mental health following the extension of emergency measures nationwide and the differential effects of containment measures across groups of respondents. Before 9 March 2020, the mental health score, which was 78.5 (95% CI 77.3–79.6) in the entire sample, varied significantly across gender and age groups, with females (77.3, 95% CI 75.8–78.8) and older individuals, namely, those aged 55+ (74.7, 95% CI 72.4–77.0 for 55–69 years old; 74.9, 95% CI 71.5–78.4 for 70+ years old), reporting the lowest scores. Regarding household structure and composition, respondents cohabiting with a partner had a comparatively higher mental health score (81.2, 95% CI 79.9–82.6). Turning to socioeconomic variables, individuals who were employed had higher scores (80.1, 95% CI 78.5–81.7) than those who were unemployed (72.9, 95% CI 61.2–66.4).

**Table 3. Random-effects regression analysis showing the change in mental health scores associated with the implementation of COVID-19 lockdown measures.**

| | Pre-lockdown (95% CI) | | Post-lockdown (95% CI) | | Pre-Post average change (95% CI) | | Cohen's d (Effect size) | p-value | Wald test p-value |
|---|---|---|---|---|---|---|---|---|---|
| Total sample | 78.5 | (77.3; 79.6) | 67.9 | (66.5; 69.4) | -10.5 | (-12.1; -9.0) | -.445 | < .001 | |
| Age | | | | | | | | | |
| 16–34 | 84.2 | (81.3; 87.1) | 66.5 | (62.7; 70.3) | -17.7* | (-21.8; -13.6) | -.744 | < .001 | < .001 |
| 35–44 | 81.6 | (78.7; 84.6) | 67.6 | (63.8; 71.4) | -14.1* | (-18.1; -10.0) | -.621 | < .001 | |
| 45–54 | 80.2 | (77.7; 82.7) | 68.4 | (65.0; 71.7) | -11.8* | (-15.5; -8.2) | -.502 | < .001 | |
| 55–69 | 74.7 | (72.4; 77.0) | 67.3 | (64.4; 70.1) | -7.4* | (-10.4; -4.5) | -.316 | < .001 | |
| 70 or more | 74.9 | (71.5; 78.4) | 71.9 | (67.8; 76.1) | -3.0 | (-7.5; 1.5) | -.136 | .186 | |
| Living with a partner | | | | | | | | | |
| No | 75.2 | (73.2; 77.2) | 66.4 | (64.0; 68.8) | -8.8* | (-11.4; -6.2) | -.358 | < .001 | .093 |
| Yes | 81.2 | (79.9; 82.6) | 69.4 | (67.5; 71.4) | -11.8* | (-13.8; -9.7) | -.515 | < .001 | |
| Testing for COVID-19 | | | | | | | | | |
| No | 78.6 | (77.4; 79.8) | 67.9 | (66.4; 69.4) | -10.7* | (-12.3; -9.1) | -.461 | < .001 | .544 |
| Yes | 78.8 | (75.9; 81.7) | 69.9 | (65.0; 74.8) | -8.9* | (-14.4; -3.5) | -.331 | .001 | |
| Increase in mortality rate at the municipal level | | | | | | | | | |
| Up to 10% | 77.1 | (75.4; 78.7) | 67.5 | (65.5; 69.5) | -9.6* | (-11.7; -7.4) | -.402 | < .001 | .473 |
| 11–50% | 79.6 | (77.8; 81.4) | 68.0 | (65.6; 70.4) | -11.6* | (-14.2; -8.9) | -.481 | < .001 | |
| 51% or more | 83.2 | (80.3; 86.2) | 71.7 | (67.2; 76.3) | -11.5* | (-15.9; -7.0) | -.543 | < .001 | |
| Sex | | | | | | | | | |
| Male | 80.6 | (78.8; 82.4) | 72.5 | (70.3; 74.8) | -8.1* | (-10.6; -5.6) | -.354 | < .001 | .018 |
| Female | 77.3 | (75.8; 78.8) | 65.3 | (63.3; 67.2) | -12.1* | (-14.1; -10.0) | -.504 | < .001 | |
| Age of the youngest child | | | | | | | | | |
| No child aged 0–14 | 78.9 | (77.7; 80.2) | 68.5 | (66.9; 70.1) | -10.4* | (-12.2; -8.7) | -.439 | < .001 | .027 |
| 0–6 years | 78.9 | (74.9; 82.9) | 62.8 | (57.0; 68.5) | -16.2* | (-22.2; -10.2) | -.678 | < .001 | |
| 7–14 years | 75.8 | (71.4; 80.2) | 70.0 | (64.9; 75.1) | -5.8 | (-11.0; -0.6) | -.271 | < .030 | |
| Education° | | | | | | | | | |
| Up to lower secondary | 78.2 | (75.9; 80.4) | 68.6 | (66.0; 71.3) | -9.5* | (-12.6; -6.5) | -0.383 | <0.001 | 0.236 |
| Upper secondary | 79.4 | (77.9; 90.0) | 69.4 | (67.3; 71.5) | -10.1* | (-12.2; -7.9) | -0.437 | <0.001 | |
| Tertiary | 77.1 | (74.3; 80.0) | 64.0 | (60.3; 67.3) | -13.4* | (-17.0; -9.8) | -0.596 | <0.001 | |
| Employment status° | | | | | | | | | |
| Employed | 80.1 | (78.5; 81.7) | 69.7 | (67.5; 71.9) | -10.4* | (-12.8; -8.1) | -0.466 | <0.001 | 0.652 |
| Unemployed | 72.9 | (67.1; 78.7) | 61.6 | (55.8; 67.4) | -11.3* | (-17.3; -5.3) | -0.461 | <0.001 | |
| Economically inactive | 77.1 | (74.2; 80.1) | 68.8 | (65.1; 72.4) | -8.4* | (-12.6; -4.1) | -0.343 | <0.001 | |
| Retired | 78.3 | (75.2; 80.5) | 66.5 | (62.8; 70.1) | -11.9* | (-15.9; -7.9) | -0.474 | <0.001 | |
| Shortage of living space° | | | | | | | | | |
| No | 79.3 | (78.1; 80.4) | 68.8 | (67.3; 70.3) | -10.5* | (-12.1; -8.9) | -0.451 | <0.001 | 0.921 |
| Yes | 73.2 | (68.4; 78.0) | 62.4 | (57.8; 67.0) | -10.8* | (-16.6; -5.0) | -0.399 | <0.001 | |

° Variables measured in ITA.LI wave 1.

* Statistically significant at the 5% level after the Bonferroni correction for multiple testing in Model 2.

Between April and September 2020 (i.e. at the time of the ITA.LI COVID-19 interviews), the mental health score was 67.9 (66.5–69.4) for the entire sample, which indicated a significant deterioration (β = -10.5). Cohen's d (-.445) suggested that the effect size was medium. Hence, Hypothesis 1 is supported.

The findings of the multiple-interaction model revealed that the mental health of the younger age groups was the most severely affected. For those aged 16–34 years, the estimated mental health score was -17.7 points (p < .001, Cohen's d = -.744) lower than the pre-lockdown

baseline measurement. The mental health score was also -14.1 points lower in people aged 35–44 years (p < .001, Cohen's *d* = -.621); however, those aged 55–69 years experienced a relatively small reduction of -7.4 points (p < .001, Cohen's *d* = -.316). For individuals aged 70+ no statistically significant change in mental health scores was recorded. For individuals living without a partner, those tested because had symptoms or were potentially exposed to the virus, and people living in municipalities with high rates of new coronavirus infections the COVID-19 crisis failed to bring about statistically significant worsening mental health. Hence, the social isolation hypothesis (Hypothesis 2) was confirmed only with respect to age.

Furthermore, comparisons of estimated marginal means showed that downward trends in mental health scores were significantly more pronounced for women (β = -12.1, *p* < .001, Cohen's *d* = -.504) than for men (β = -8.1, *p* < .001, Cohen's *d* = -.354), and for those cohabiting with and taking care of children aged 0–6 years (β = -16.2, *p* < .001, Cohen's *d* = -.678) than for individuals in households with no children below the age of 14 (β = -10.4, *p* < .001, Cohen's *d* = -.439). Accordingly, the heavier workload hypothesis (Hypothesis 3) was fully sustained.

For any of the remaining subgroups—that is the lower educated, the unemployed, and people with poorer housing conditions—pre- and post-lockdown estimated mental health scores were not significantly different from each other at a 5% significance level. Therefore, the socioeconomic resource hypothesis (Hypothesis 4) was rejected.

Seasonal sensitivity analyses on the heterogeneity of COVID-related effects provided similar results (see Table 1 in S1 File).

## Discussion and conclusion

To the best of our knowledge, this study is the first to use a subsample of a household panel study to track changes in mental health in Italy before the onset of the COVID-19 pandemic and into the first lockdown period, from late April to early September 2020. The results of RE regression analyses fitted on a sample (N = 897) of respondents aged 16 years and older provided support for our main research hypothesis and indicated that there was a substantial and statistically significant short-term deterioration in mental health, as measured by a composite index derived from the mental component of the SF-12.

In addition, we found evidence of significant heterogeneity in the COVID-related effects on mental health. In particular, parameter estimates proved that negative changes in mental health were unevenly distributed across the sample by age. Indeed, findings confirm previous research [3, 15, 17] and suggest that, at least during the first wave of the pandemic, while older age groups were the most infected and faced the greatest risk of severe illness and death—the younger age group (namely, those aged 16–34 years) was the most affected in terms of mental distress. Moreover, this study provides further support for previous studies that pointed to both the gender-specific effects of COVID-19, with women suffering more than men the mental health consequences of the outbreak in the short-term, and to a steeper decline in mental health from pre-pandemic baseline levels for individuals with pre-schoolers, who were exposed to more stressful childcare dynamics [3, 10, 15, 16].

There are several possible explanations for these results. The RE models showed that differences in employment or marital status, direct or indirect exposure to infection and perception of a shortage of living space did not impact the summary measure of mental health. Hence a plausible explanation for the worst effects on mental health found in younger age groups is linked to the COVID-induced reduction in social relations. In other words, the social distancing requirements and policy-induced variation in early life caused by responses to the COVID-19 outbreak produced harsher consequences for those aged 16–34 years, who, before

the pandemic had enjoyed more frequent and intense social relationships outside of the home —that is, at school, at work, at the neighbourhood level, and so on. Another possible explanation is that young people were the most deeply concerned about their future—because of crushed employment opportunities, possible financial hardship, or fears over not being able to have children and form stable families—or were the most exposed to the fear of or to the actual course of illness and death of close family members or friends [34]. These were all variables that could not be controlled for in the models. When assessing the short-term differential effects of the COVID-19 pandemic on mental health, we also found that women and individuals living with and taking care of preschool-age children were disproportionately affected, consistent with the workload research hypothesis. It seems possible that these results were due to the over-representation of female employment in jobs and sectors that have been at the forefront of the COVID-19 response, such as health care, but also teaching and retailing, which were profoundly impacted by the physical environment, work intensity, and working time quality [35–37]. It worth mentioning that women have been the main component of the workforce in the hardest-hit sectors, such as accommodation and food service activities, which may have resulted in higher perceived job insecurity or even greater job and income losses, which may have brought about increased mental distress [35]. That said, in Italy, gross imbalances in the household distribution of unpaid care work remain; thus, the closure of childcare services put an additional burden and strain on women, especially on those who working remotely [38, 39]. Multiple sources reveal an intensified risk of gender-based domestic violence, harassment, and abuse in times of lockdown and quarantine [40], possibly resulting in substantial mental health consequences.

In sum, we may argue that the COVID-19 outbreak has, on the one hand, been levelling the social gradient in mental health, as long as it has translated into disadvantages for young people, who had previously enjoyed greater psychological well-being but, on the other hand, the pandemic has widened the mental health gender gap and also created new inequalities, based on the age of the youngest child being taken care of within the household. As stated before, these findings may be interpreted as immediate and unintended short-term consequences of containment and mitigation policies [31]. However, these effects may persist in the longer run, as the direct and indirect social and economic consequences of COVID-19 gradually unfold [41], and the available research suggests that the levels of mental distress are higher than expected even once the lockdown has eased and mitigation policies are adapted to new circumstances [16]. Therefore, it is important that future research tracks and models change in mental health in the population, as well as in specific subgroups, relative to baseline levels measured before and during the COVID-19 pandemic. Furthermore, future research and policy-making on containment and mitigation measures should consider the differential effects found in this study and should shed light on the indirect impact of the disruption of normal daily activities and not just on the net impact of government lockdown measures in terms of economic and employment contraction. COVID-related consequences, and the effects pertinent to COVID-induced mitigation policies, are likely to include more than just financial strain and labour market vulnerability. The unintended and possible outcomes deserve specific attention. For instance, isolation and loneliness for younger people have been found to result in subsequent health risk behaviours [42] and adverse effects in terms of sense of purpose, ability to make decisions, and feeling of having a meaningful and useful role to play in life [17]. In this regard, in Italy, which is among the most affected countries worldwide, the immediate worries about the COVID-19 emergency and economic crisis have added to the fears and concerns that the young adults have been suffering the most (i.e. unemployment, precarious work, life uncertainty, etc.). This may result in harsh, cumulative, and longer-term troubles in young people's lives: facing the pandemic at a crucial stage of the life course—in which professional

careers and social identities are shaped and decisions on forming a new household and starting a family are taken—may further delay the transition to adulthood, as life plans and priorities have been severely hit [22]. The same holds true for women and individuals with pre-school age children, whose overall well-being and life chances have already been undermined by the gender-biased and familistic nature of the Mediterranean model of the welfare state [43].

As previously discussed, there was no significant evidence confirming previous research that pointed to the differential effects across socioeconomic groups [3, 10, 15, 16]. Indeed, the magnitude of the COVID-related outcomes associated with differences in employment status, educational attainment, and housing conditions was not statistically significant. These unexpected results may be related to the limited sample size, which may hamper the possibility of detecting changes in mental health for specific subgroups; additionally, they may be related to the fact that only past values of socioeconomic variables were used as explanatory variables in the models, which did not allow amounts of recent past to be brought into the prediction. A further limitation, which concerns period and maturation threats to the validity of the one-group pre-test–post-test design used in this study [44], needs to be acknowledged. Specifically, we cannot exclude that mental changes occurred within the respondents following regular seasonal patterns [e.g., 45, 46], which could account for the results. However, sensitivity analysis confirmed the patterns of the results (see Table 1 in S1 File). Nonetheless, we could not account for other potential trends in mental distress that had already occurred, regardless of the pandemic. In this regard, the lack of multiple pre-pandemic measurements did not allow us to conduct further analysis and forced us to assume *a priori* the absence of maturation threats to the internal validity of our one-group pre-test–post-test design. However, recent longitudinal studies that were able to draw on several waves of data collection before and during the pandemic detected higher-than-expected reductions in mental health scores; therefore, the trajectories of change in mental health scores at times of COVID-19 were different from previous trends [15, 17]. Finally, there might have been a differential non-response to the ITA.LI COVID-19 survey, which could lead to biased parameter estimates, namely to an overestimation of possible negative COVID-related effects to mental health.

## Supporting information

**S1 File.**
(DOCX)

## Author Contributions

**Conceptualization:** Marta Pancheva, Egidio Riva.

**Formal analysis:** Mario Lucchini, Tiziano Gerosa, Maurizio Pisati.

**Methodology:** Mario Lucchini, Tiziano Gerosa, Maurizio Pisati, Chiara Respi.

**Writing – original draft:** Tiziano Gerosa, Marta Pancheva, Chiara Respi, Egidio Riva.

**Writing – review & editing:** Egidio Riva.

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
