## [Decision Letter · Decision Letter 0]

21 Jul 2021

PONE-D-21-13037

The differential effect of Covid-19 pandemic on the mental health of young and healthy persons: Evidence from ITA.LI - Italian Lives, the Italian household panel

PLOS ONE

Dear Dr. Riva,

Thank you for submitting your manuscript to PLOS ONE. After careful consideration, we feel that it has merit but does not fully meet PLOS ONE’s publication criteria as it currently stands. Therefore, we invite you to submit a revised version of the manuscript that addresses the points raised during the review process.

We look forward to receiving your revised manuscript.

Kind regards,

Stéphanie Baggio

Academic Editor

PLOS ONE

Journal Requirements:

Additional Editor Comments:

In addition to the relevant comments of the reviewers that should be adressed, please:

- add the baseline response rate and retention rate and, if relevant, comparisons for baseline variables between responders and non-responders at follow-up

- remove formula from the subsection Statistical analyses (not needed for the understanding of our readers)

Reviewers' comments:

Reviewer's Responses to Questions

**Comments to the Author**

1. Is the manuscript technically sound, and do the data support the conclusions?

Reviewer #1: Partly

Reviewer #2: Partly

2. Has the statistical analysis been performed appropriately and rigorously? 

Reviewer #1: N/A

Reviewer #2: No

3. Have the authors made all data underlying the findings in their manuscript fully available?

Reviewer #1: Yes

Reviewer #2: No

4. Is the manuscript presented in an intelligible fashion and written in standard English?

Reviewer #1: No

Reviewer #2: Yes

5. Review Comments to the Author

Reviewer #1: The authors here present a study investigating self-perceived mental health in a group of Italian people before and after the Covid-19 lockdown. They only include data on self-perceived mental health from 904 participants who answered the SF-12 questionnaire both prior to March 9 2020 and again during lockdown between April and September 2020. With their data they highlight a general increase in mental distress, which is found to impact the younger population and those without prior health conditions more.

With their longitudinal approach, and data dating back prior to the Covid-19 pandemic, the data is generally quite interesting; however, I do have some major concerns regarding the analyses and reporting of the data. Unfortunately, the manuscript is not very well written and it would be recommended to make the language clearer and more understandable. It is often difficult to follow the points that the authors are trying to make, and the sentences are often very long. The section on statistical analyses needs to be revised in order for the readers to more easily follow the methods behind the findings.

Specific comments:

- Abstract: needs revision, particularly regarding presenting the results but also for concluding remarks

- Abstract, line 30: Both in the abstract and in the discussion the authors state that they have included 906 participants; however, throughout the manuscript it is clear that the actual number is 904.

- line 31: “indicate that there was a significant”??? is it significant or not? And provide the estimates

- line 33-37: estimates are lacking and are the effects small or large?

- Introduction, line 69: Please insert relevant references when stating that you ”draw on available evidence from previous pandemics”.

- Introduction, line 70-78: Please specify what is your primary and secondary hypotheses and outcomes

- There are many tests – the secondary outcomes should preferable undergo multiple testing

- Table 1: Some variables herein are not easy to understand and needs some rephrasing; Please change the word gender to sex (relevant throughout the manuscript), change ”testing of symptoms” to ”testing for Covid-19”?, note that mortality rates are for the municipalities of the individuals, and either remove the asterix* after Employment status or explain the asterix below the table.

- Table 1: It is not clear what the “mean score” is off and this is not defined in the table. There should also be p-values to see if the difference is significant and preferable relative risk estimates with CI also to evaluate the magnitude of the differences

- Table 2: why is the results for the total sample not adjusted?

- Variables, line 133: Please specify what is meant by ”economically inactive”.

- Variables, line 135: I would prefer a rephrasing of ”shortage of space in the dwelling” to something along the lines of ”shortage of living space”, which would make it easier to understand.

- Variables, line 135-136: Please in your manuscript specify how data on previous health conditions were obtained, and optimally also which conditions are included herein.

- Findings: I would find it highly relevant to be able to read somewhere the p-values for the different observations of differences in mental health between subgroups as reported on in the first part of the findings sections. This could perhaps be added to Table 1.

- Findings, line 199: The sentence ”which was indicative of a possible overall increase in mental stress” does not correspond with the observation of a significant overall decrease in mental health scores with a p-value below 0.001 Please rephrase.

- Regarding analyses performed in the manuscript I find it highly relevant to investigate the following; 1) can the difference in self-perceived mental health between males and females be explained by age? And the same question applies for educational level, previous health conditions and taking care of children between 0-14 years. All of these variables are most likely highly influenced by age. 2) Can you stratify previous health conditions in to physical and mental health conditions and analyze them separately?

- Table 2: Please make sure that you use the same labels for the variables in both Table 1 and Table 2.

- Table 2, legend: Are you here mentioning that data on employment status and education level is obtained only post-lockdown? This needs to be highlighted previously in the manuscript, and should also be discussed as a limitation. In particular employment status might have changed with the lockdown, and so your analyses of mental health in subgroups of employment status prior to the lockdown renders less useful.

- Findings: You do not mention the findings of your sensitivity analyses anywhere in your findings section.

Reviewer #2: Dear Authors

This paper investigates changes in mental health between the period before COVID-19 and the COVID-19 period and finds that there was an overall decrease in mental health, which was most marked in young people and those without prior health conditions.

The paper is overall well written. However, some details need to be clarified, and I have some questions regarding multiple points.

1.Title: “differential” effect really means “higher” effect here and could maybe be written as such.

2.Abstract: Please mention the data collection period of the pre covid wave in the abstract

3.Line 86: This is confusing: the data collection period stopped in March 2020 for the purpose of this study as later respondents were not invited. The next date mentioned is April 2020, which is the start of data collection. The date of 9 March 2020 should be mentioned earlier and clearer as the end of the data collection period of wave 1.

4.Please clarify why no ethic statement is required. In my opinion, this type of study requires an ethical statement, including the name of the ethical review board and a mention of how participants consent was obtained. Collection of health related data with the possibility to link records to earlier data collection waves and to e-mail addresses/phone number, i.e. non-anonymous data collection clearly requires ethical approval from an ethical review board/commission at least in my country (Switzerland). I was under the impression that this is very similar in the European Union. If you did not obtain ethical approval and/or, please provide evidence that this was not needed under Italian law. Please also mention which identifier was used to link the data (line 99).

5.Line 105: Please specify which translation from the SF-12 was used. It would also appear that no reference refers specifically to the SF-12, the only reference refers to an evaluation paper for the SF-36.

6.The coding instructions for the SF-12 https://www.researchgate.net/publication/242636950_SF-12_How_to_Score_the_SF-12_Physical_and_Mental_Health_Summary_Scales are to use factor score weights normed to the population for the mental and physical health scores, with all 12 items loading on both factors. You used a simple sum score measure of only the 6 items for mental health, respectively your two factor model (but based on only 6 items). Please explain why you did not use the scoring method as described by the authors. There may be good reason for that (different population etc), but this should be clarified.

7.Line 115: I do not understand what the two factors were? Just reading like this it would seem that there was a factor for wave 1 and one for wave 2, while the point of such an analysis is usually to measure the same factor across time. Please reformulate and clarify this.

8.Line 125: in the title of S1 Fig, an “of” is missing after “distribution”.

9.Line 124: Please cite a reference for the MAP method. It is very unclear where the summary measure mentioned in line 106 were used and were the factor scores.

10.Line 125: How were the scores normalized? With respect to the sample or with respect to the reference values of the SF-12 coding instructions (https://www.researchgate.net/publication/242636950_SF-12_How_to_Score_the_SF-12_Physical_and_Mental_Health_Summary_Scales)

11.Please mention the software used in your analysis

12.Line 281: There are multiple points to consider regarding the findings about vulnerable groups. First, what is the role of floor effects? Can those with already low mental health even decrease or are they at the bottom floor and can only increase? And could regression to the mean explain why those with better mental health decrease more (towards the mean) than those with lower mental health (who could rather be expected to increase towards the mean)? If I understand that right, you only tested for overall heterogeneity of mean differences across a predictor (i.e. age) and did not contrast changes between groups and adjust for individual pre-crisis mental health, therefore your analysis did likely not account for these effects. Thus, it is possible that the between group differences are entirely due to regression to the mean (essentially measurement error) rather than the effect of the crisis. So for example in line 226, it follows from your analysis that those with no underlying health condition decreased in mental health, while those with a mental health condition did not decrease. However, I see no analysis showing that the effect of the crisis is greater in those without underlying health condition, compared to those with an underlying health condition, when taking baseline levels into account.

13.Related to the underlying health condition group, unless only chronic health problems were measured, most health related problems tend to improve over a period of several month, what is often accompanied with an improvement in mental health, whether there was a crisis in between or not. This alternative explanation for the increasing score / not decreasing score may be worth considering.

14.As regards the adjusted analysis, I do not see the rationale for basing the discussion exclusively on the adjusted results. Lets take employment status as an example: Those unemployed would appear to decrease more in mental health than those retired. If you adjust this for age (which probably had the most impact because of the large differences according to age), there is naturally no more difference between the groups, because those retired are older than the unemployed. Does this mean that there is no effect of unemployment or retirement on changes in mental health? In my opinion, no. There may be an effect of unemployment or retirement, which is however logically confounded with age. Similarly, having children between 0-6 years is naturally unlikely in those 16 years old and in those older than 50, and will be most frequent at the age between 25 and 40. What is the rationale of adjusting such a variable for age? That those with children between 0-6 years showed a higher decrease in mental health would be of some interest, and the fact that they were of a certain age does not invalidate that result. In my opinion, most of the discussion about the crisis as “leveller of mental health” is based on a questionable adjustment strategy. Judging from the raw scores it clearly looks like the mental health of those unemployed (before the crisis) deteriorated more compared to those employed, although this was not tested statistically in your paper.

15.Again regarding vulnerable groups, I wonder why on of the most vulnerable group, those with low mental health before the crisis (i.e. those with a low SF-12 score), were not considered and investigated.

16.Line 287: should distress be “decrease” ?

17.Line 257: I see no evidence that the sharper decrease in young people is due to social restrictions. It may as well have been due to fear of for example financial hardship, fear of losing once older relatives, etc, variables you did not adjust for.

18.Line 307: If, the levelling only concerned the gradient in mental health, not health in general.

6. PLOS authors have the option to publish the peer review history of their article (what does this mean?). If published, this will include your full peer review and any attached files.

Reviewer #1: No

Reviewer #2: No

---

## [Author Response · Author response to Decision Letter 0]

4 Sep 2021

Dear editor and reviewers,

we would like to thank you for your comments and advice, which we believe helped us a lot in improving the overall clarity, soundness, and readability of the paper. The revised version of our study is attached. We made several major changes to the original version, as it is also highlighted in the marked-up copy of the manuscript, which is attached. Responses to each point raised during the review process are listed below. Briefly, based on your comments, suggestions, and concerns, which have all been addressed, we:

- better framed the main argument and spelt out more clearly our research hypotheses 

- performed more appropriately and rigorously the statistical analyses

- presented and discussed more in depth the results of the study. 

Please note that the English has been thoroughly and professionally reviewed and meets the required standards for publication, as well as PLOS ONE’s style requirements, including those for file naming.

Thank you. 

Reviewer 1

Reviewer: The authors here present a study investigating self-perceived mental health in a group of Italian people before and after the Covid-19 lockdown. They only include data on self-perceived mental health from 904 participants who answered the SF-12 questionnaire both prior to March 9, 2020, and again during lockdown between April and September 2020. With their data they highlight a general increase in mental distress, which is found to impact the younger population and those without prior health conditions more. With their longitudinal approach, and data dating back prior to the Covid-19 pandemic, the data is generally quite interesting; however, I do have some major concerns regarding the analyses and reporting of the data. Unfortunately, the manuscript is not very well written and it would be recommended to make the language clearer and more understandable. It is often difficult to follow the points that the authors are trying to make, and the sentences are often very long. The section on statistical analyses needs to be revised in order for the readers to more easily follow the methods behind the findings.

Reviewer: Abstract: needs revision, particularly regarding presenting the results but also for concluding remarks

The abstract was revised accordingly

- line 30: Both in the abstract and in the discussion the authors state that they have included 906 participants; however, throughout the manuscript it is clear that the actual number is 904.

Thank you. That was a typo. Following your request, we better controlled and adjusted the size of the final sample used in our analysis (N = 897). A detailed description of the sample selection process is provided in the data section. 

- line 31: “indicate that there was a significant”??? is it significant or not? And provide the estimates

- line 33-37: estimates are lacking and are the effects small or large?

We revised that sentence and provided parameter estimates and added the effect sizes (Cohen’s d) 

- Introduction, line 69: Please insert relevant references when stating that you draw on available evidence from previous pandemic.

We revised that sentence and provided additional references, as requested 

- Introduction, line 70-78: Please specify what is your primary and secondary hypotheses and outcomes

We believe we better framed the study and spell out clearly our main research hypotheses

- There are many tests – the secondary outcomes should preferable undergo multiple testing

We would like to thank you for advice. We used the Bonferroni correction for multiple testing in Model 2 (i.e. the model estimating multiple-interactions between time predictor and covariates). Furthermore, we added the unadjusted p-values for each parameter estimate.

- Table 1: Some variables herein are not easy to understand and needs some rephrasing; Please change the word gender to sex (relevant throughout the manuscript), change ”testing of symptoms” to ”testing for Covid-19”?, note that mortality rates are for the municipalities of the individuals, and either remove the asterix* after Employment status or explain the asterix below the table.

We revised the text accordingly. 

- Table 1: It is not clear what the “mean score” is off and this is not defined in the table. There should also be p-values to see if the difference is significant and preferable relative risk estimates with CI also to evaluate the magnitude of the differences

We opted to report mean pre-post values of the dependent variable and its average variation over time in Table 2 (Results of the RE regression models). In this table, as you requested, we added the p-values for all the estimates. Moreover, to evaluate the magnitude of the variations over time, we computed Cohen’s d effect size for single-group pre-post design with continuous outcomes.

- Table 2: why is the results for the total sample not adjusted?

Please note that, based on the suggestion and advice that the second reviewer provided, we changed the analytical strategy. As documented in the paper, first we estimated a random effect model with pre- and post-lockdown period indicators as the only predictors of variation in mental health. Subsequently, the interactions between the pre- and post-Covid lockdown period indicators and the entire set of explanatory variables were included. We abandoned the two-stage strategy based on testing single interaction (labelled “unadjusted” in the previous version of the paper) models before the multiple-interaction one (labelled “adjusted”). Consequently, the issue of “not adjusted” and “adjusted” models no longer stands. 

- Variables, line 133: Please specify what is meant by ”economically inactive”.

We revised the text accordingly. 

- Variables, line 135: I would prefer a rephrasing of ”shortage of space in the dwelling” to something along the lines of ”shortage of living space”, which would make it easier to understand.

We revised the text accordingly. 

- Variables, line 135-136: Please in your manuscript specify how data on previous health conditions were obtained, and optimally also which conditions are included herein.

Based on your advice, we reflected on the specificity of this predictor and concluded that it was too vague (as for the definition of the type and chronicity of the underlying health condition). Moreover, it could semantically overlay with the outcome. For these reasons, we decided to remove it from the analyses.

- Findings: I would find it highly relevant to be able to read somewhere the p-values for the different observations of differences in mental health between subgroups as reported on in the first part of the findings sections. This could perhaps be added to Table 1.

We revised the text accordingly. As for p-values please see what we have already indicated in previous response to comments. 

- Findings, line 199: The sentence ”which was indicative of a possible overall increase in mental stress” does not correspond with the observation of a significant overall decrease in mental health scores with a p-value below 0.001 Please rephrase.

It was a typo. We meant increase in mental distress, instead. We revised the text accordingly. 

- Regarding analyses performed in the manuscript I find it highly relevant to investigate the following; 1) can the difference in self-perceived mental health between males and females be explained by age? And the same question applies for educational level, previous health conditions and taking care of children between 0-14 years. All of these variables are most likely highly influenced by age. 2) Can you stratify previous health conditions in to physical and mental health conditions and analyze them separately?

We would like to thank you for your advice. As for the point n.1, please note that in our study – both in the original version and in the revised version – we focussed on general research hypotheses on the Covid-related effects, and building on available research, we selected a few independent variables (listed in the introduction). Testing for gender-specific differences related to age, as you suggest, is no doubt a relevant research proposition, which we believe is suitable for a different paper. However, due to the small size of the sample, we believe that introducing in the model multi-way interactions across levels of covariates and the time predictor could result in a concrete risk of over-parameterization. Turning to issue raised in point 2, we are sorry but, unfortunately, we cannot do what you invite us to do. We have at our disposal only measures of mental health. However, following your advice, we additionally checked the multi-group stability of the pre-post variations in mental health over the distribution of the outcome with a quantile regression approach, and the results show a high degree of stability.

- Table 2: Please make sure that you use the same labels for the variables in both Table 1 and Table 2.

The table was revised accordingly. 

- Table 2, legend: Are you here mentioning that data on employment status and education level is obtained only post-lockdown? This needs to be highlighted previously in the manuscript, and should also be discussed as a limitation. In particular employment status might have changed with the lockdown, and so your analyses of mental health in subgroups of employment status prior to the lockdown renders less useful.

Employment status was included in the models as a lagged variable in ITA.LI wave 1. That was discussed as a possible limitation of the study in the revised version of the paper. 

- Findings: You do not mention the findings of your sensitivity analyses anywhere in your findings section.

Findings of sensitivity analyses are now mentioned in the findings section and also reported in the Annex

 

Reviewer 2: 

Dear Authors

This paper investigates changes in mental health between the period before COVID-19 and the COVID-19 period and finds that there was an overall decrease in mental health, which was most marked in young people and those without prior health conditions. The paper is overall well written. However, some details need to be clarified, and I have some questions regarding multiple points.

1.Title: “differential” effect really means “higher” effect here and could maybe be written as such.

2.Abstract: Please mention the data collection period of the pre covid wave in the abstract

The title of the paper and the abstract were revised based on both your suggestions and the results of the analyses, which were run differently following the points raised by reviewer #1. 

3.Line 86: This is confusing: the data collection period stopped in March 2020 for the purpose of this study as later respondents were not invited. The next date mentioned is April 2020, which is the start of data collection. The date of 9 March 2020 should be mentioned earlier and clearer as the end of the data collection period of wave 1.

We are sorry for confusion. We now made it clear that data collection for ITA.LI wave 1 started in June 2019 and ended in December 2020, while the ITA.LI COVID-19 survey was conducted from 20 April to 2 September 2020. We also clearly stated that the final sample for this study did not include respondents who completed ITA.LI wave 1after 9 March 2020. 

4.Please clarify why no ethic statement is required. In my opinion, this type of study requires an ethical statement, including the name of the ethical review board and a mention of how participants consent was obtained. Collection of health related data with the possibility to link records to earlier data collection waves and to e-mail addresses/phone number, i.e. non-anonymous data collection clearly requires ethical approval from an ethical review board/commission at least in my country (Switzerland). I was under the impression that this is very similar in the European Union. If you did not obtain ethical approval and/or, please provide evidence that this was not needed under Italian law. Please also mention which identifier was used to link the data (line 99).

ITA.LI – Italian Lives data collection protocols were written in accordance with data protection law (Regulation EU 2016/679, Italian Legislative Decree 196/2003) and were approved by the Ethics Committee of the University of Milano-Bicocca (protocol number 0042665/19). Therefore, data collection obtained an ethical approval and observed specific laws and rules. As for sample participant consent, at the time of the first meeting with the interviewer, potential respondents were given the information sheet (with detailed info about personal data treatment) and the consent form. If the respondents agreed to take part in research, they had to fill in the research consent form with their personal details. Concerning the identifier, respondents were assigned a unique random code, which was used to merge data from the ITA.LI wave 1 and ITA.LI COVID-19 survey

5.Line 105: Please specify which translation from the SF-12 was used. It would also appear that no reference refers specifically to the SF-12, the only reference refers to an evaluation paper for the SF-36.

Thank you for your advice. We provided additional reference for the Italian translation of the SF-12. It is a study that cross-validate it in 9 countries, including Italy. See Gandek et al. [18]

6.The coding instructions for the SF-12 https://www.researchgate.net/publication/242636950_SF-12_How_to_Score_the_SF-12_Physical_and_Mental_Health_Summary_Scales are to use factor score weights normed to the population for the mental and physical health scores, with all 12 items loading on both factors. You used a simple sum score measure of only the 6 items for mental health, respectively your two factor model (but based on only 6 items). Please explain why you did not use the scoring method as described by the authors. There may be good reason for that (different population etc), but this should be clarified.

7.Line 115: I do not understand what the two factors were? Just reading like this it would seem that there was a factor for wave 1 and one for wave 2, while the point of such an analysis is usually to measure the same factor across time. Please reformulate and clarify this.

Mental health was measured in ITA.LI wave 1 (i.e. pre-lockdown) using the 12 items generally used in the Short-Form of the Health Survey (SF-12), which was adapted to the Italian population. As you suggested, scores from the 12 items are computed using standardized scoring algorithms to construct physical and mental component summary measures (PCS-12 and MCS-12). However, the ITA.LI COVID-19 survey questionnaire (i.e. post-lockdown), due to time/budget constraints, could collect only 6 items of the SF-12; specifically those assessing the mental component of the SF-12 scale. Hence, for the scope of this study we could only use the 6 items assessing mental health, which were asked twice (i.e. before and after the introduction of lockdown measures) to the respondents. Accordingly, we validated a different summary measure of mental health, based on the analysis of pooled data and principal components analysis. This approach, which could also address the points that you raised in the revision process, produced a composite measure of mental health that – as results of convergent validity analysis suggest – was strongly correlated with the mental component of SF-12 computed using the standard US algorithm. More details are provided in the revised version of the paper. 

8.Line 125: in the title of S1 Fig, an “of” is missing after “distribution”.

That was a typo. We revised the text accordingly.

9.Line 124: Please cite a reference for the MAP method. It is very unclear where the summary measure mentioned in line 106 were used and were the factor scores.

As just explained, in the revised version of the paper we used a different scoring method based on a factor weights and regression approach.

10.Line 125: How were the scores normalized? With respect to the sample or with respect to the reference values of the SF-12 coding instructions (https://www.researchgate.net/publication/242636950_SF-12_How_to_Score_the_SF-12_Physical_and_Mental_Health_Summary_Scales)

We normalized the scores with respect to the sample, since we validated a new version of the scale (6 items) which could not be directly compared with the mental health component of the SF-12 (12 items).

11.Please mention the software used in your analysis

We used STATA 17. It is now mentioned at the end of the “Analytic strategy” section 

12.Line 281: There are multiple points to consider regarding the findings about vulnerable groups. First, what is the role of floor effects? Can those with already low mental health even decrease or are they at the bottom floor and can only increase? And could regression to the mean explain why those with better mental health decrease more (towards the mean) than those with lower mental health (who could rather be expected to increase towards the mean)? If I understand that right, you only tested for overall heterogeneity of mean differences across a predictor (i.e. age) and did not contrast changes between groups and adjust for individual pre-crisis mental health, therefore your analysis did likely not account for these effects. Thus, it is possible that the between group differences are entirely due to regression to the mean (essentially measurement error) rather than the effect of the crisis. So for example in line 226, it follows from your analysis that those with no underlying health condition decreased in mental health, while those with a mental health condition did not decrease. However, I see no analysis showing that the effect of the crisis is greater in those without underlying health condition, compared to those with an underlying health condition, when taking baseline levels into account.

We had already discussed the possibility of a “floor-effect” that when writing the original version of the paper. To address this point, in the revised version of the study we additionally checked the multi-group stability of the pre-post variations in mental health over the distribution of the outcome with a quantile regression approach. Results show a high degree of stability; thus, we believe there should not be such a risk. 

13.Related to the underlying health condition group, unless only chronic health problems were measured, most health related problems tend to improve over a period of several month, what is often accompanied with an improvement in mental health, whether there was a crisis in between or not. This alternative explanation for the increasing score / not decreasing score may be worth considering.

As already indicated (see comments to reviewer #1), we considered the specificity of the items assessing underlying health conditions. After careful consideration, we believe that its formulation was too vague to understand what the nature and actual extent of underlying health conditions were. Accordingly, following the advice of reviewer #1, we decided not to include this item in the models. 

14.As regards the adjusted analysis, I do not see the rationale for basing the discussion exclusively on the adjusted results. Lets take employment status as an example: Those unemployed would appear to decrease more in mental health than those retired. If you adjust this for age (which probably had the most impact because of the large differences according to age), there is naturally no more difference between the groups, because those retired are older than the unemployed. Does this mean that there is no effect of unemployment or retirement on changes in mental health? In my opinion, no. There may be an effect of unemployment or retirement, which is however logically confounded with age. Similarly, having children between 0-6 years is naturally unlikely in those 16 years old and in those older than 50, and will be most frequent at the age between 25 and 40. What is the rationale of adjusting such a variable for age? That those with children between 0-6 years showed a higher decrease in mental health would be of some interest, and the fact that they were of a certain age does not invalidate that result. In my opinion, most of the discussion about the crisis as “leveller of mental health” is based on a questionable adjustment strategy. Judging from the raw scores it clearly looks like the mental health of those unemployed (before the crisis) deteriorated more compared to those employed, although this was not tested statistically in your paper.

We decided to abandon the two-stage strategy based on the adjustment approach. In the revised version of the paper, we first estimated a random-effects model (RE) with pre- and post-lockdown period indicator as the only predictor of change in mental health scores. In a second RE model the interactions between the pre- and post-lockdown period indicator and the entire set of explanatory variables were included. Based on the results of the Hausman test, we rejected the hypothesis that the individual-level effects could be more adequately modelled by a fixed-effects model. 

15.Again regarding vulnerable groups, I wonder why on of the most vulnerable group, those with low mental health before the crisis (i.e. those with a low SF-12 score), were not considered and investigated.

To check the stability of mental health score variations overtime conditional to its distribution before the lockdown, we ran a quantile regression. Analyses confirmed the stability of the coefficients across quantiles.

16.Line 287: should distress be “decrease” ?

It was a typo. 

17.Line 257: I see no evidence that the sharper decrease in young people is due to social restrictions. It may as well have been due to fear of for example financial hardship, fear of losing once older relatives, etc, variables you did not adjust for.

We revised the discussion and conclusion accordingly. 

18.Line 307: If, the levelling only concerned the gradient in mental health, not health in general.

It was a typo. 

Additional Editor Comments

In addition to the relevant comments of the reviewers that should be addressed, please:

- add the baseline response rate and retention rate and, if relevant, comparisons for baseline variables between responders and non-responders at follow-up

As indicated in the revised version of the paper, 950 of 2,415 eligible people (i.e. respondents who took part in ITA.LI wave 1 and for whom an email and/or a phone number was/were available and valid) participated in the ITA.LI COVID-19 survey [30]. Hence, the final response rate of the ITA.LI COVID-19 survey was 39.3% (AAPOR response rate 1). The final response rate of the ITA.LI wave 1 survey was 37.5% (AAPOR response rate 1). Retention rates for ITA.LI survey will be calculated over wave 2, which is projected to start in fall 2021. 

We believe that providing descriptive statistics, for all the variables included in the models, for both respondents and non-respondents is not relevant. Please note that, in the revised version of this study, after careful consideration we decided not to use inverse probability weights, which adjusted for differential non response to the ITA.LI COVID-19 survey. This is due to the following reasons: i) results from the subsample of ITA.LI COVID-19 survey could not be generalized to the all sample of ITA.LI wave 1 respondents; and ii) results of weighted regression analysis were pretty similar to those of unadjusted regression analysis 

- remove formula from the subsection Statistical analyses (not needed for the understanding of our readers)

We removed the formula from the paper.

---

## [Decision Letter · Decision Letter 1]

18 Oct 2021

PONE-D-21-13037R1Differential effects of COVID-19 and containment measures on mental health: Evidence from ITA.LI - Italian Lives, the Italian household panelPLOS ONE

Dear Dr. Riva,

Thank you for submitting your manuscript to PLOS ONE. After careful consideration, we feel that it has merit but does not fully meet PLOS ONE’s publication criteria as it currently stands. Therefore, we invite you to submit a revised version of the manuscript that addresses the points raised during the review process.

We look forward to receiving your revised manuscript.

Kind regards,

Stéphanie Baggio

Academic Editor

PLOS ONE

Reviewers' comments:

Reviewer's Responses to Questions

**Comments to the Author**

1. If the authors have adequately addressed your comments raised in a previous round of review and you feel that this manuscript is now acceptable for publication, you may indicate that here to bypass the “Comments to the Author” section, enter your conflict of interest statement in the “Confidential to Editor” section, and submit your "Accept" recommendation.

Reviewer #1: (No Response)

Reviewer #2: All comments have been addressed

2. Is the manuscript technically sound, and do the data support the conclusions?

Reviewer #1: Partly

Reviewer #2: Yes

3. Has the statistical analysis been performed appropriately and rigorously? 

Reviewer #1: I Don't Know

Reviewer #2: Yes

4. Have the authors made all data underlying the findings in their manuscript fully available?

Reviewer #1: No

Reviewer #2: No

5. Is the manuscript presented in an intelligible fashion and written in standard English?

Reviewer #1: Yes

Reviewer #2: Yes

6. Review Comments to the Author

Reviewer #1: - Abstract: The relative and absolute differences are still not clear – it is important to the readers that it is clear if it is a small but significant effect

- It is also important to mention in the abstract that there was no significant changes for the group >70 years and that it was most pronounced among the youngest group 16-34 years of age were there were a 21% absolute decline on the score

- It should be highlighted that there might be a bias of the individuals attending here for the second time, which was under the lockdown, and they might be biased towards more likelihood of people responding that felt affected by the pandemic. So this should also be clear with how many in total took part of the first investigation, and how many of these also responded here the second time, also in the abstract. As the attendance seemed to be low for this follow-up investigation, which could be a considerable bias

- And good that the authors sorted out the number of participants, which ended up to be different than the two numbers initially reported

- I don’t really see how the authors see that hypothesis 2 was really fulfilled, as there are many outcomes in hust this hypothesis, where most where not significant. So if their hypothesis 2 were just if there were any differences in any subgroups, these should also be adjusted for multiple testing regarding all the subgroups included in this

- How was hypothesis 3 sustained?

- It might be easier to read if they have subheadings in the result section for each of the hypothesis investigated

- Overall, I'm also concerned how much their used questions actually represent the mental health. Results for specific questions are not presented, and the overall score seem to also have questions as "accomplished less" and "limited social activities", which a lockdown will induce for most, but it doesn't answer how their mental health is affected, and you can score significantly lower on these items for instance, but still have the same overall mental health. Like the question "feeling downhearted and low" is a bit closer to what the article tries to answer, so how is the difference on this question for instance?

Reviewer #2: Dear Authors,

Thank you for your careful consideration of my comments. I am overall happy with your replies to my comments. I have some very minor comments:

-The abbreviation RE and FE are not introduced in the paper.

-In table 3 in the last column, it should be mentioned that this is the p-value of the Wald test

7. PLOS authors have the option to publish the peer review history of their article (what does this mean?). If published, this will include your full peer review and any attached files.

Reviewer #1: No

Reviewer #2: No

---

## [Author Response · Author response to Decision Letter 1]

22 Oct 2021

Rebuttal letter 

Dear editor and reviewers, please find listed below the responses to each point raised during the review process. Responses are in italics. 

Reviewer 1

- Abstract: The relative and absolute differences are still not clear – it is important to the readers that it is clear if it is a small but significant effect

- It is also important to mention in the abstract that there was no significant changes for the group >70 years and that it was most pronounced among the youngest group 16-34 years of age were there were a 21% absolute decline on the score

We revised the abstract accordingly 

- It should be highlighted that there might be a bias of the individuals attending here for the second time, which was under the lockdown, and they might be biased towards more likelihood of people responding that felt affected by the pandemic. So this should also be clear with how many in total took part of the first investigation, and how many of these also responded here the second time, also in the abstract. As the attendance seemed to be low for this follow-up investigation, which could be a considerable bias

We clearly acknowledged this limitation in the final section of the paper. Please note that due to word limit count we could not add all this information in the abstract 

- And good that the authors sorted out the number of participants, which ended up to be different than the two numbers initially reported

We would like to thank you for your comments and notes to previous round of reviews, which helped in sorting this issue out. 

- I don’t really see how the authors see that hypothesis 2 was really fulfilled, as there are many outcomes in hust this hypothesis, where most where not significant. So if their hypothesis 2 were just if there were any differences in any subgroups, these should also be adjusted for multiple testing regarding all the subgroups included in this

We revised the discussion of the results accordingly. In particular, building on Kuhn and colleagues we hypothesized (Hypothesis 2) a stronger deterioration of mental health among those more exposed to the detrimental effects of social isolation, such as young adults, individuals living without a partner, and people belonging to a COVID-19 risk group (i.e. those tested because had symptoms or were potentially exposed to the virus and people living in municipalities with high rates of new coronavirus infections) (Hypothesis 2). In the revised version of the study we clearly stated, in a separate paragraph (to make it clearer), as follows:

“The findings of the multiple-interaction model revealed that the mental health of the younger age groups was the most severely affected. For those aged 16–34 years, the estimated mental health score was -17.7 points (p < .001, Cohen’s d = -.744) lower than the pre-lockdown baseline measurement. The mental health score was also -14.1 points lower in people aged 35–44 years (p < .001, Cohen’s d = -.621); however, those aged 55–69 years experienced a relatively small reduction of -7.4 points (p < .001, Cohen’s d = -.316). For individuals aged 70+ no statistically significant change in mental health scores was recorded. For individuals living without a partner, those tested because had symptoms or were potentially exposed to the virus, and people living in municipalities with high rates of new coronavirus infections the COVID-19 crisis failed to bring about statistically significant worsening mental health. Hence, the social isolation hypothesis (Hypothesis 2) was confirmed only with respect to age.”

Please note that in the previous version of this study we already stated that Hypothesis 2 was only partially confirmed, just with respect to age. Besides, we had already run multiple testing analyses in previous round of review, too. 

- How was hypothesis 3 sustained?

We revised the discussion of the results accordingly. In particular, again, building on Kuhn and colleagues, as well as on previous research conducted during the pandemic, we hypothesized (Hypothesis 3) a stronger deterioration of mental health among individuals with a heavier workload, such as women and people with preschool-age children in the household (Hypothesis 3). In the revised version of the paper, we discussed the results as follows: 

“Furthermore, comparisons of estimated marginal means showed that downward trends in mental health scores were significantly more pronounced for women (β = -12.1, p < .001, Cohen’s d = -.504) than for men (β = -8.1, p < .001, Cohen’s d = -.354), and for those cohabiting with and taking care of children aged 0–6 years (β = -16.2, p < .001, Cohen’s d = -.678) than for individuals in households with no children below the age of 14 (β = -10.4, p < .001, Cohen’s d = -.439). Accordingly, the heavier workload hypothesis (Hypothesis 3) was fully sustained.”

- It might be easier to read if they have subheadings in the result section for each of the hypothesis investigated

In third-round review, we tested the hypotheses and presented the findings in separate paragraphs, which we believe improved the clarity and readability of the paper. 

- Overall, I'm also concerned how much their used questions actually represent the mental health. Results for specific questions are not presented, and the overall score seem to also have questions as "accomplished less" and "limited social activities", which a lockdown will induce for most, but it doesn't answer how their mental health is affected, and you can score significantly lower on these items for instance, but still have the same overall mental health. Like the question "feeling downhearted and low" is a bit closer to what the article tries to answer, so how is the difference on this question for instance?

The 12-item Short Form Survey (SF-12) is a general health questionnaire that was first published in the mid-90s as part of the Medical Outcomes Study (MOS). It is one of the most widely used measure of mental health. See, for instance: Ware J, Kosinski M, Keller SD. A 12-item short-form health survey: construction of scales and preliminary tests of reliability and validity. Med Care. 1996; 34:220–233. doi: 10.1097/00005650-199603000-00003. Hence, it has been extensively studied and employed over time. In this study we had already provided additional evidence of the reliability and robustness of the mental health-related scale that includes the selected items. Furthermore, in response to your concerns, please find below some additional analyses, which further support the scale. 

Variable Frequencies (%) p-value

During the past 4 weeks, have you had any of the following problems with your work or other regular daily activities as a result of any emotional problems (such as feeling depressed or anxious)?

Have you accomplished less than you would like?

 Yes No 

Pre 93 (11) 765 (89) - - - - <0.001*

Post 250 (29) 608 (71) - - - - 

Didn't do work or other activities as carefully as usual?

 Yes No 

Pre 86 (10) 772 (90) - - - - <0.001*

Post 246 (29) 612 (71) - - - - 

These questions are about how you feel and how things have been with you during the past 4 weeks. For each question, please give the answer that comes closest to the way you have been feeling. How much of the time during the past 4 weeks…

Have you felt downhearted and low?

 All of the time Most of the time A good bit of the time Some of the time A little of the time None of the time 

Pre 7 (1) 29 (3) 29 (3) 163 (19) 422 (49) 208 (24) <0.001¥

Post 18 (2) 54 (6) 49 (6) 261 (30) 321 (37) 155 (18) 

Did you have a lot of energy?

 None of the time A little of the time Some of the time A good bit of the time Most of the time All of the time 

Pre 9 (1) 48 (6) 218 (25) 186 (22) 280 (33) 117 (14) 0.006¥

Post 12 (1) 89 (10) 262 (31) 90 (11) 304 (35) 101 (12) 

Have you felt calm and peaceful?

 None of the time A little of the time Some of the time A good bit of the time Most of the time All of the time 

Pre 8 (1) 36 (4) 162 (19) 183 (21) 329 (38) 140 (16) 0.001¥

Post 10 (1) 63 (7) 227 (27) 101 (12) 344 (40) 113 (13) 

During the past 4 weeks, how much of the time has your physical health or emotional problems interfered with your social activities?

 All of the time Most of the time Some of the time A little of the time None of the time 

Pre 6 (1) 29 (3) 100 (12) 202 (24) 521 (61) - <0.001¥

Post 17 (2) 60 (7) 215 (25) 311 (36) 255 (29) - 

* p-values derived from the Mc Nemar’s test for dichotomous variables

¥ p-values derived from the Wilcoxon signed rank sum test for ordered categorical variables

 

Reviewer #2: Dear Authors, Thank you for your careful consideration of my comments. I am overall happy with your replies to my comments. I have some very minor comments:

-The abbreviation RE and FE are not introduced in the paper

We revised the text accordingly

-In table 3 in the last column, it should be mentioned that this is the p-value of the Wald test

We revised the text accordingly

---

## [Editor Report · Decision Letter 2]

2 Nov 2021

Differential effects of COVID-19 and containment measures on mental health: Evidence from ITA.LI - Italian Lives, the Italian household panel

PONE-D-21-13037R2

Dear Dr. Riva,

We’re pleased to inform you that your manuscript has been judged scientifically suitable for publication and will be formally accepted for publication once it meets all outstanding technical requirements.

Kind regards,

Stéphanie Baggio

Academic Editor

PLOS ONE
---

## [Editor Report · Acceptance letter]

5 Nov 2021

PONE-D-21-13037R2 

Differential effects of COVID-19 and containment measures on mental health: Evidence from ITA.LI - Italian Lives, the Italian household panel 

Dear Dr. Riva:

I'm pleased to inform you that your manuscript has been deemed suitable for publication in PLOS ONE. Congratulations! Your manuscript is now with our production department. 

Kind regards, 

on behalf of

Dr. Stéphanie Baggio 

Academic Editor

PLOS ONE